# Using VGI and Social Media Data to Understand Urban Green Space: A Narrative Literature Review

**Nan Cui** *, **Nick Malleson**, **Victoria Houlden and Alexis Comber**

School of Geography, University of Leeds, Leeds LS2 9JT, UK; n.s.malleson@leeds.ac.uk (N.M.);
V.Houlden@leeds.ac.uk (V.H.); A.Comber@leeds.ac.uk (A.C.)
*  Correspondence: gync@leeds.ac.uk; Tel.: +44-7821023959

**Abstract:** Volunteered Geographical Information (VGI) and social media can provide information about real-time perceptions, attitudes and behaviours in urban green space (UGS). This paper reviews the use of VGI and social media data in research examining UGS. The current state of the art is described through the analysis of 177 papers to (1) summarise the characteristics and usage of data from different platforms, (2) provide an overview of the research topics using such data sources, and (3) characterise the research approaches based on data pre-processing, data quality assessment and improvement, data analysis and modelling. A number of important limitations and priorities for future research are identified. The limitations include issues of data acquisition and representativeness, data quality, as well as differences across social media platforms in different study areas such as urban and rural areas. The research priorities include a focus on investigating factors related to physical activities in UGS areas, urban park use and accessibility, the use of data from multiple sources and, where appropriate, making more effective use of personal information. In addition, analysis approaches can be extended to examine the network suggested by social media posts that are shared, re-posted or reacted to and by being combined with textual, image and geographical data to extract more representative information for UGS analysis.

**Keywords:** urban green space; volunteered geographical information; social media data

## 1. Introduction

Urban green space (UGS) refers to urban land covered by vegetation [1]. It is an essential component of urban environmental systems and plays a critical role in sustaining urban natural environments as well as the social systems that use these spaces [2].An increasing number of studies have examined the various benefits of UGS to humans via the interactions between humans and UGS [3]. These include studies of the ecosystem services of UGS [4], the events and physical activities that occur in UGS areas [5,6], the benefits to mental health [2,7], and the accessibility of UGS [8,9]. These studies have confirmed that city residents largely rely on parks and green spaces for physical, mental, and social well-being [10,11]. UGS is therefore recognised as one of the key features supporting urban sustainability and enhancing the quality of life of urban residents [12].

Worldwide, the proportion of people living in urban areas will increase from 50% in 2010 to nearly 70% by 2050 [13]. Hence, the demand for UGS is rapidly increasing in the context of urbanisation, especially in metropolitan areas. This means that the planning and management of UGS is critical in order to satisfy the needs of urban residents [14], requiring urban planners to make public places more liveable and sustainable [15]. The interactions between humans and UGS, in particular, play a fundamental role in UGS planning [16]. For example, researchers have investigated the interactions between UGS and humans and their impacts on visitors' perception, as well as the benefits to residents' well-being [17,18].

Social media are internet-based applications that enable people to communicate and share resources [19]. These technologies allow the public to voluntarily produce geographic

information which can be considered as Volunteered Geographic Information (VGI). The georeferenced data provided by social media can be considered as VGI and social media as VGI sources. Examples of this are geotagged Tweets from Twitter, geotagged photographs from Flickr and Instagram, etc. [20]. VGI is defined as user-generated digital geographical data, including both text and multimedia [21], enabled through the use of a range of technologies to create, assemble, and disseminate geographic information. VGI can be used to support the understanding and exploration of the socio-economic and environmental conditions of a place through the analysis of different resources such as geotagged Tweets and photos [22,23], check-in data [24], OpenStreetMap [25], etc. The widespread use of popular social media technologies such as Twitter, Facebook, Instagram and Flickr where users post and share their views, opinions, feelings and emotions provides a resource to examine UGS visits, behaviours and use [26]. For example, studies have investigated perceptions of green environment quality by analysing park visit frequency through Point-of-Interest (PoI) check-ins [27,28], mapping cultural service areas [29,30] and investigating tourism patterns [31,32]. Such data potentially provide opportunities for researchers to quickly obtain a large amount of useful information for scientific research [33].

This review covers the use of major social media data platforms in urban green space research and examines data collection methods, the advantages and disadvantages of different social media VGI and highlights a number of research gaps. It does this by considering the following questions:

- What were the research aims and the research topics in studies that explored VGI in relation to urban green space?
- What types of social media websites or platforms were generally selected in these studies?
- What were the methods used in collecting data, processing data and analysing data?
- What were the potential challenges and problems not yet resolved and researched?

The reason for this review now, focussed in this way, is because previous reviews about the application of VGI data in urban studies have mainly focused on smart city planning and management [34,35], data acquisition and quality issues [36], data mining approaches and techniques [21,37], and human mobility in urban areas [38], with a focus on the broader context of urban management and planning [39,40]. However, in the domain of UGS and VGI data application, few reviews have summarised the application of VGI data in the context of UGS planning.

## 2. Materials and Methods

In this study, a bibliometric analysis of published research was undertaken in order to support investigation of the characteristics of previous studies (Section 3). Then, the key research areas (themes) were examined as well as the methods used (including data pre-processing as well as spatial, temporal and semantic analysis) before highlighting a number of data quality issues and key areas for methodological improvement.

### 2.1. Bibliometric Literature Search

A bibliometric analysis was undertaken using 4 steps (Figure 1) based on established guidelines for conducting a systematic literature review [3,41]. The aims of this analysis were to first establish the degree to which UGS analyses are increasingly using different forms of social media to understand UGS user attitudes and preferences, and then to determine the how they were being used (for example, in support of specific objectives such as tourism or ecosystem services benefits). This review examined articles published between 1 January 2010 and 1 December 2019 in English. First, the search terms were determined based on a number of keywords, which can be classified into two groups. One group was composed of words related to "urban green space" [42]. The other group referred to "social media" or "volunteered geographic information". The search terms are described in Table 1 and relate to two themes: topic (e.g., urban green space) and data sources (e.g., social media data). These were adapted for each database to ensure

appropriate syntax. The search terms in this review were selected based on the authors' knowledge and previous studies examining methods to conduct a systematic review [3,42]. The search engines Web of Science, Scopus, IEEE Xplore and Google search were used as they cover a range of discipline areas, with the aim of capturing all relevant literature in this domain. The search terms were used to find matches in "title, abstract, and keywords" for Scopus and "Topic" for Web of Science. A final step was to synthesise the data and to extract relevant information.

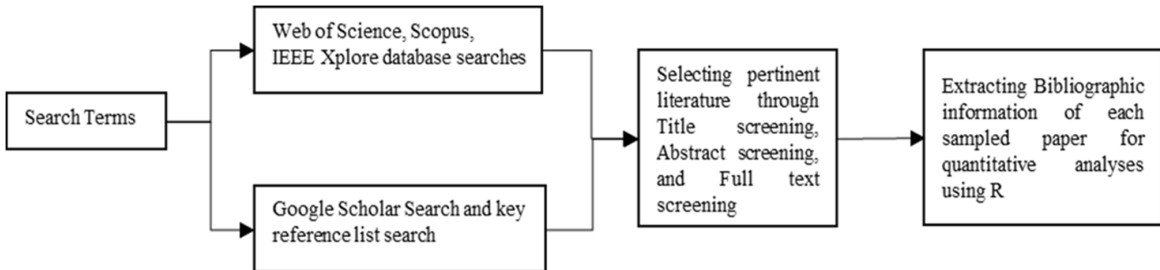

**Figure 1.** Outline of search strategy.

**Table 1.** Summary of literature search terms and their use in the search query.

| UGS | | | Data | |
|---|---|---|---|---|
| Urban | AND | Green space/Greenspace | AND | Social media |
| | OR | Green infrastructure | OR | Volunteer geographic information/ VGI |
| | OR | Park | OR | Crowd sourced geographic information |
| | OR | Recreation area | OR | Crowd source/Crowdsource/Crowdsourcing |
| | OR | Garden | OR | Citizen science/Citizen contributed science |
| | OR | Playing field | OR | Flikr/Twitter/Weibo/Foursquare/Instagram |
| | | | OR | WeChat/WhatsApp/Facebook |

The query for paper selection by key words was (TITLE-ABS-KEY ("urban") AND ("green space" OR "greenspace" OR "green infrastructure" OR "park" OR "recreation area" OR "Garden" OR" playing field ") AND TITLE-ABS-KEY("Social media" OR "Volunteer geographic information" OR "VGI" OR "crowd source" OR "citizen science" OR "Flickr" OR "Twitter" OR "Weibo" OR "Foursquare" OR "Instagram")) AND PUBYEAR > 2009 AND PUBYEAR < 2020.

After entering the search terms in each database, the papers were screened and some of them were excluded according to the content of the title or abstract. This was to remove articles that were not related or only marginally related to the objectives of the review. For example, articles examining the use of social media data without urban green space visitation were excluded. In addition, the literature considered in this review was restricted to publications in international, peer-reviewed journal articles and conference proceedings. The remaining papers were further screened for the exclusion criteria in Table 2. In addition, papers that did not appear in the initial search results but were referenced within the identified papers were included if they related to the review aims [42]. Finally, the bibliographic information of each paper was extracted for quantitative analyses, including trend detection, text and topic mining, and citation analysis. A final manual check of the papers was undertaken to ensure a minimum equal evaluation of topics and themes and as little assessment bias as possible [3].

**Table 2.** Literature screening exclusion criteria.

| No. | Exclusion Criteria | Examples |
|:---:|:---:|:---:|
| 1 | Studies not written in English | [43] |
| 2 | Studies concerned with intelligent parking systems | [44,45] |
| 3 | Studies concerned with app information monitoring | [46] |
| 4 | Surveillance of health by using web data | [47] |
| 5 | Studies not related to green space | [48] |
| 6 | Studies that selected industrial parks as study areas | [49] |
| 7 | Studies concerned with disaster detection | [50] |
| 9 | Studies concerned with emergency situations | [51] |

### 2.2. Data Processing

Bibliometric methods allow researchers to examine, organise, and analyse huge amounts of information to find hidden patterns [52]. Many bibliometric tools use information about authors, affiliations and citations to identify and explore patterns in conceptual maps, co-citation analyses, cluster and factor analyses [53]. The *"bibliometrix"* R package (http://www.bibliometrix.org) (accessed on 3 March 2021) [54], an open-source tool for scientometric and bibliometric research, was used for quantitative analysis and for topic mining of the bibliographic data in R 4.0.3 (https://cran.r-project.org/bin/windows/base/old/4.0.3/) (accessed on 3 March 2021). This package includes all major bibliometric analysis methods, with rapid analysis speeds and the use of data matrices for co-citation, coupling, collaborative analysis, and co-word analysis. In this study, *bibliometrix* was used to extract information such as annual publication rates, corresponding authors' country, country scientific production (i.e., countries of author affiliations), conceptual structure maps and cumulative occurrence of keywords. A co-word analysis was undertaken using the *bibliometrix* R-package to undertake multiple correspondence analysis (MCA) to examine the conceptual structure of the domain [54]. MCA is an exploratory multivariate technique for the graphical and numerical analysis of multivariate categorical data [55]. In the co-word analysis undertaken here, the words are plotted on a two-dimensional map.

### 3. Results

#### 3.1. Main Characteristics of Included Studies

The total number of articles identified from the database search was 802. Screening the papers based on the exclusion criteria (Table 2) resulted in 219 articles, and 177 articles remained after reading the full texts and analysing each article individually. Details of the volume of generated papers and the originating countries of their authors are shown in Figure 2.

The number of documents published per year in Figure 2a indicates that the number of papers has increased continuously since 2010, entering a more rapid growth phase in 2014. This demonstrates that scholars have increasingly studied UGS by using social media data in recent years, or that social media has become more popular. Additionally, Wi-Fi infrastructure may have been improved, with local managers providing Wi-Fi within UGS areas, making it easier to obtain data for research. The increasing number of papers indicates the increasing significance of UGS. Figure 2b shows the number of corresponding authors' country and the degree of international collaboration is through the proportions that are associated with single country publications (SCP) and with multiple country publications (MCP). The United States has the largest total number of publications, followed by China, Spain, the United Kingdom and Australia. Additionally, Finland and the UK have the greatest proportion of MCP, followed by Portugal and Denmark, suggesting that these counties have higher levels of international collaboration than others.

Figure 3a shows the clustering of the topics identified from the *author-specified* keywords. This was generated by a Multiple Correspondence Analysis (MCA) of the topics. MCA allows researchers to study the association between two or more nominal categorical data [56], and this approach can be used to understand the fields of selected papers from a



low-dimensional perspective. Specifically, the nearer the positions of the points, the closer the concepts are that they indicate. Figure 3a shows that four clusters of co-words exist.

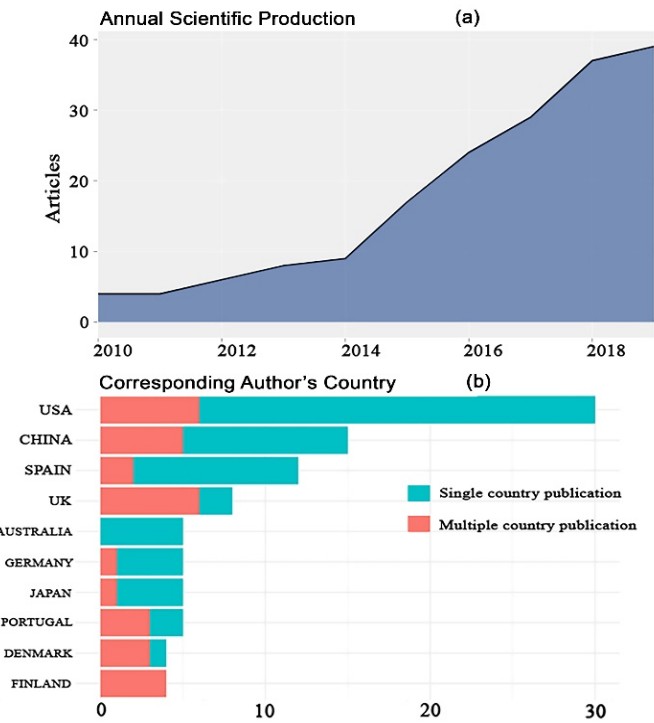

**Figure 2.** Bibliometric analyses of UGS and social media research: (**a**) Annual scientific production, (**b**) Corresponding author's country.

Cluster 1 includes words related to urban green space and environment. This shows that the focus of papers was mainly centred on urban areas and green space. In addition, the words related to geographic information systems (GIS) and sentiment analysis were identified as common research methods and analysis tools in this cluster, indicating that these approaches made great contributions in the field. Cluster 2 includes themes related to ecosystem services, tourism, urban planning and behaviour research. Additionally, Twitter, Instagram, Flickr, and OpenStreetMap were also included in this cluster, indicating that these social media platforms were selected as the main data sources in this field. In this case, Figure 3a shows that Twitter data are closer to ecosystem services and travel behaviour in this map. This shows that Twitter was a popular data source in this area of research; Flickr and OpenStreetMap are closer to human mobility and tourism, which shows that these sources were more popular in these areas of research in relation to UGS. Social media analysis, urban parks and green space were observed in Cluster 3, indicating that social media can be used as a new resource in the analysis of urban parks. Ecosystem system services were found in Cluster 4, indicating the focus on urban parks as the main source of natural landscapes to provide important ecosystem services for urban residents. This map helps researchers to understand existing research themes in the analysis of UGS by using VGI and social media data, and which data platforms were more popular in which research themes. Figure 3b shows the cumulative occurrence of the keywords in all 177 articles. The highest numbers of keywords are social media, followed by Twitter, big data, cultural ecosystem services, Flickr and tourism, which indicates that these areas may be important research topics in relation to the studies of VGI data and UGS.

Overall, Figure 3 shows that the keywords and abstract terms in the selected articles mainly concentrated on ecosystem services, human behaviour, urban planning and tourism by using various social media data related to urban green space and urban parks. This is not a surprise given the search terms of this review; however, the words about physical activities in UGS areas and factors related to urban park use and the accessibility of urban

green space did not appear in these clusters. This is a potential area for future research, as discussed in Section 4.

**Figure 3.** (**a**) Conceptual structure map and (**b**) cumulative occurrence of the keywords in the UGS and social media literature.

### 3.2. Data Sources in Relation to UGS Analysis

The data sources used in UGS research were summarised from all reviewed articles by scanning the section "data resources" in each paper. In addition, data acquisition approaches including data collection websites, software and data platform availability were also recorded and summarised in Table 3. The advantages and disadvantages of the top five popular data platforms are highlighted below. Additionally, in order to understand why certain types of data sources were selected by authors when they studied different themes, the "introduction" section was summarised to find more detailed descriptions of data sources from the authors' perspective. Figure 4 shows the frequency of different data platforms used in the 177 articles over different years. It shows that, overall, social media data including Twitter, Flickr, Instagram and Weibo are becoming increasingly popular in studies relating to UGS, and the data platforms of Twitter and Flickr are the most frequently used as data sources. Twitter is a very popular microblogging service established in 2006. Twitter users "tweet" about their individual opinions and feelings within a 140-character (now 280) limit [57]. Flickr was established in 2004 and is the most popular online photo management and sharing application in the world [58]. Instagram, established in 2010, is used to share self- and user-generated content [59]. Weibo is a large social network website

in China. Weibo users can obtain up-to-date status information, provide status updates, share views, and communicate with others [60].

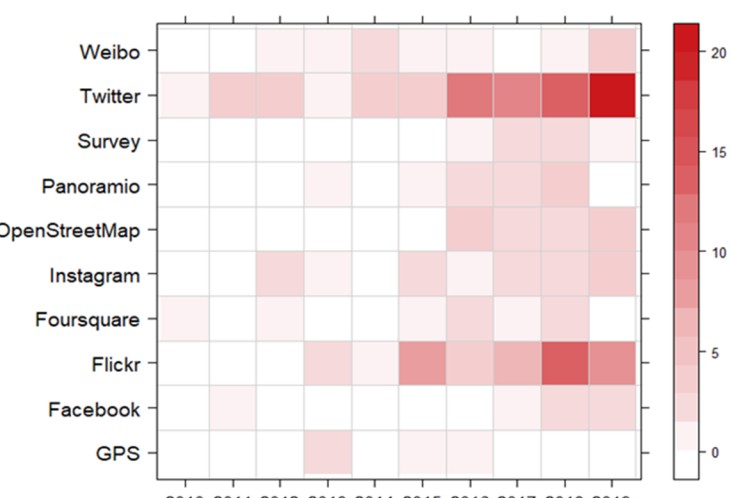

**Figure 4.** The frequency of occurrence of different data platforms found in the UGS and social media literature.

Twitter was selected as the data platform by 71 articles, accounting for 39% of all papers, which indicates that this data platform was the most popular in the research works related to UGS, followed by Flickr (40), Instagram (10), Weibo (9) and OpenStreetMap (9). Other, less well known, VGI platforms included MapMyFitness [61], Tencent [62], Tuniu [63], Wikiloc [61] and Wikipedia.

The social media platforms identified in this review were classified into three categories according to [64]: text-based social media such as Twitter, Weibo; image-based social media such as Flickr, Instagram; map-based social media such as MapMyFitness, Baidu and Google Maps.

Text-based social media data have been mainly used to investigate park visitation [65–67], factors affecting park use [24,61,68], physical activity and events in park areas [69–71], and the emotional response of visitors in park areas [68,72,73]. The reasons why text-based data were popular in these research topics can be summarised as follows:

1. The data are easy to collect using methods such as public application programming interfaces (APIs), such as Twitter streaming APIs and Weibo APIs (Table 3), and can be downloaded at as frequent a time interval as necessary [16,39].
2. There are large numbers of users on these networks, generating huge amounts of information [24,61,68].
3. The georeferenced text-based social media data allow researchers to investigate park visitation patterns from a spatial perspective, while achieving greater longitudinal depth [65–67].
4. The time of text-based data (i.e., Tweets) creation can support investigations into the temporal patterns of park visitation [74].
5. The content of text data can be used in semantic analysis including sentiment analysis and emotion detection, which can help scholars understand the public perceptions and interest in urban green space areas [72,73].

Image-based social media data (such as Instagram and Flickr) were mainly used in research examining cultural ecosystem services [75–77], park visitation [65], investigations of factors affecting park use [78], and physical activities [79] for the following reasons:

1. The photographs that social media users post may reflect their interests, aesthetic values, sentimental attachment and emotional state at a particular time and place [75–77].

2. Georeferenced photos allow researchers to detect spatial patterns of park visitation and user behaviour [65]. User profiles help researchers identify where visitors live and their home location [79].
3. Shared pictures provide access to real-time information, allowing researchers to generate temporal patterns of urban green space use [79]. Additionally, images are taken and posted throughout the year, enabling longitudinal analysis.
4. These platforms provide free, up-to-date, and high spatial and temporal resolution information sources [32,80].

There are some limitations associated with social media data that the papers discuss. These include low coverage, data quality, uncertainties, and problems with representativeness and reliability [39,72,81]. In addition, existing analysis methods for information extraction need to be improved [82]. These limitations should not be ignored by researchers. For example, in research examining spatiotemporal park visit patterns using semantic information from Twitter, researchers are often faced with data-specific uncertainties, including identifying the locational information of visitors, which affects the nature of the information extracted [82]. In addition, Twitter users only represent a small proportion of the real park visitor population; users are usually younger, wealthier and have more educational qualifications as compared to the general population [83,84]. This has been an ongoing concern for many of the papers reviewed. Thus, the use of geo-social media data such as georeferenced photos and geo-Tweets should not replace the consideration of traditional methods when it comes to the assessment of urban park visitation [74,75]. However, georeferenced Tweets or photos still have the potential to produce valuable and useful knowledge, particularly in metropolitan areas with a high density of social media users [72].

Research should always consider the validity of social media data before analysing them in order to determine the extent to which the results robustly support management and planning. For example, Lenormand et al. [85] validated the use of Twitter data in Barcelona and Madrid by comparing different data sources including the census and cell phone data. The results showed that the three data sources provided comparable information for studies of urban human mobility.

Incomplete information such as uncertainty over timestamps and locations can lead to biases in UGS research. For example, the timestamps in Flickr photos can be the time the photo was taken or when it was uploaded, and geotagged locations can also be changed by users [86]. Different types of spatiotemporal analysis (such as seasonal or weekend/weekday comparison) could be affected by the uncertainty of these data [87].

Several researchers combined various datasets in order to overcome the limitations of using a single platform. For instance, some studies [65,88] used geolocated Twitter and Flickr data to explore park visitors' views and factors affecting urban park visitation. Lyu, Zhang and Greening [24] compared VGI data from Weibo and Baidu to understand the factors affecting urban park use in China. In other research [89], two VGI data sources were used, Flickr and OpenStreetMap (OSM), and then combined with remote sensing data to assess the visitation and perceived importance of UGS. The combination and comparison of different kinds of social media datasets in studies related to UGS allow researchers to generate more comprehensive conclusions about the factors associated with park visitation, UGS physical qualities and events. However, not all social media data were found to be suitable for the local context. For example, Baidu Map data were found to have more accurate location check-in information than Weibo data [24] in assessing urban parks in Wuhan, but other research was unable to establish whether Baidu Map was better in Beijing [60] and Shenzhen [90] as only Weibo data were used to assess the UGS use in these cities. This indicates a potential for bias if studies rely on a single data platform, suggesting the need to consider using a range of social media data from different platforms to enhance the reliability of the research; in other words, future works could focus on the combination of different types of social media data such as text-based data (e.g., Twitter and Weibo) and map-based data (e.g., Baidu Map and OpenStreetMap) in assessing urban park use. Table 3 summarises the characteristics of the most popular data platforms in relation to UGS studies.

**Table 3.** The social media platforms used in UGS analysis.

| Data | Platforms | | | | |
|---|---|---|---|---|---|
| | **Twitter** | **Flickr** | **Instagram** | **Weibo** | **OpenStreetMap** |
| **Data collection website** | [74,81] https://developer.twitter.com) (accessed on 3 March 2021) | [86] www.flickr.com/api (accessed on 3 March 2021) | [91] www.instagram.com/developer (accessed on 3 March 2021) | [24,90] https://open.weibo.com/development/datacenter (accessed on 3 March 2021) | [25,86] http://www.openstreetmap.org (accessed on 3 March 2021) |
| **Data type** | Text-based VGI | Image-based VGI | Image-based VGI | Text-based VGI | Map-based VGI |
| **Collection methods** | Twitter's search API, streaming API, Rest API, research API, and Twitter's Firehose [16,39]. Python wrapper. Tweepy (https://www.tweepy.org/) (accessed on 3 March 2021) python library [91]. Tweet R package [81]; TAGS Version 6.0 [92]. | Search on the Flickr developer site [32]. Using standard Hypertext Transfer Protocol (HTTP) methods to retrieve and manipulate data [93]. The Flickr API (https://www.flickr.com/services/api/) (accessed on 3 March 2021) [94]. | Using a custom-made tool written for the Python programming language [95]. Using the API of Instagram by (https://www.instagram.com/developer/) (accessed on 3 March 2021) [96]. | The location service dynamic reading interface of the Sina Weibo open platform (https://api.weibo.com/2/place/nearby/photos.json) (accessed on 3 March 2021) as the data source [66]. Data collection was facilitated by Weibo application program interfaces (APIs). Through the "to obtain nearby locations" API [90]. | QuickOSM (https://plugins.qgis.org/plugins/QuickOSM/) (accessed on 3 March 2021) Python module for QGIS was used for collecting data from OSM.The OSM data are freely downloadable from geofabrik website (http://download.geofabrik.de/asia/nepal.html) (accessed on 3 March 2021). |
| **Geography** | With geo-coordinates | Geotagged posts (including pictures, titles and text) | Geotagged posts (including pictures, titles and text) | With geo-coordinates | Active mapper communities in many locations |
| **Content** | User ID, Tweet text, timestamp, geotags and volunteered geolocations | Photo ID and owner ID, title, description, geotags, time when a photo was taken and upload time | Photo photo ID, photo title, description, tags, upload time, time when a photo was taken, location, and owner ID | Text and metadata in Weibo with geolocation, and user ID, photographs location, device type | OpenStreetMap encodes data in different formats such as points, polylines, and polygons |

**Table 3.** *Cont.*

| Data | Platforms | | | | |
|---|---|---|---|---|---|
| | **Twitter** | **Flickr** | **Instagram** | **Weibo** | **OpenStreetMap** |
| **Advantages** | Free, high spatio-temporal resolution; Lots of Twitter users post messages at various locations, including school, home, restaurants, and touristic sites. Real-time information that potentially reaches a huge audience [91]. | Free, spatially and temporally explicit, visitation hotspots. Allows for image analysis and content. User characteristic analysis, actual visitation [89]. | Online mobile application focused on sharing photographs and providing a platform for social networking [76]. | Weibo users (462 million according to the 2018 Weibo User Development Report) can upload their real-time locations and share their preferences and activities on the Internet. Data from Weibo check-ins can well represent the preferences and activities of people in urban areas [86]. | A free and up-to-date map of the world accessible and obtainable for everyone; millions of registered contributors; provides free and flexible contribution mechanisms for data (useful for map provision, routing, planning, geo-visualisation, point of interest search). Insight into people's individual perspectives and perceptions [86]. |
| **Disadvantages** | Twitter data have some biases, such as age, gender, and education. Not all the collected Tweets are usable since some of them may have been generated by spammers [97]. | Unclear meaning, confounding factors. Potential sampling and selection biases, noise in the data [93]. | Locational accuracy. The issues of anonymity and privacy arise. No information was gathered concerning the users, no socio-economic data exist, which makes it difficult to assess representability in detail [76]. | Sina Weibo check-in data have some biases, such as age, gender, a temporal change and social class bias. Weibo users are mainly composed of people between 18 and 40 years old, accounting for 89% of the total number of users. | Though OSM has no strict quality control mechanism, studies have indicated that data obtained from OSM are good enough and comparable to authoritative data to some extent [89]. |

### 3.3. Research Themes in Relation to UGS Analysis

A set of phrases were manually extracted from keywords, titles and abstracts and then ranked based on their frequency. The first 10 of these were then used to code each paper based on the occurrence or non-occurrence, as summarised in Figure 5. The themes of *cultural ecosystem services* and *urban park use* are gaining increasing attention from scholars. In detail, 44 papers researched the topic of *culture ecosystem services* provided by UGS, accounting for about 24% of all papers, making it the most popular topic. This was followed by the theme of *human–environment interactions* (36 papers), with the third most popular topic being *urban tourism* (34 papers). A total of 29 papers considered the theme of *urban park use*, 17 papers studied *environmental protection*, 7 papers focused on *human mobility patterns*, and 5 papers researched *biodiversity* and *landscape characterisation*. In relation to cultural ecosystem services in UGS, various data platforms such as Flickr, Instagram, Twitter, Panoramio [75] and Wikiloc [98] have been utilised. Amongst these platforms, Flickr was the most commonly used [95,99], whilst research examining the theme of park use has most commonly used Twitter and Weibo [24,74].

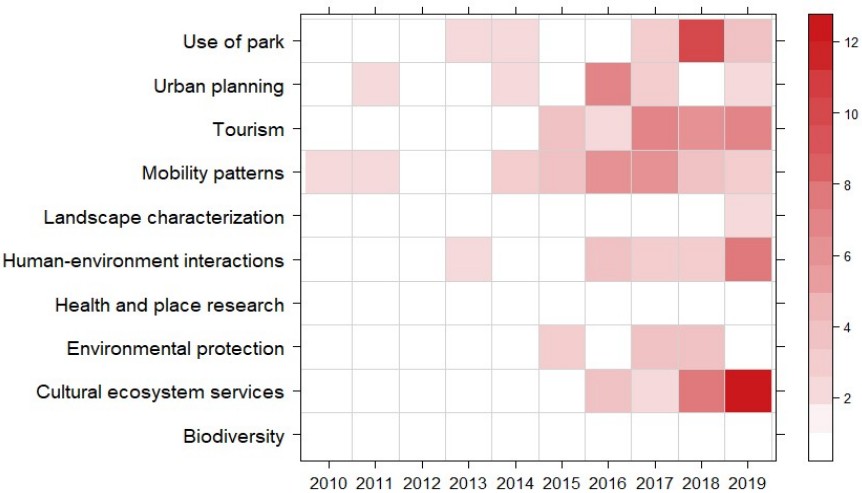

**Figure 5.** Research topics covered by research using social media and UGS.

### 3.4. Methods Used in Data Analysis

Various data and methods were used in the reviewed articles that relate to UGS studies. These have been divided these into three aspects: data pre-processing, spatial and temporal analysis and semantic analysis.

#### 3.4.1. Methods Used in Pre-Processing

A key issue is that social media data used by researchers for UGS analysis should be published by human users such as urban dwellers or tourists instead of bots or spammers [67]. Some have found that advertisers [97] and automated accounts [72] can post a huge number of messages daily or hourly, and even create geolocated messages that are posted in locations a long way from their purported location (>500 km). Such data should be identified as non-human [97] and removed.

Georeferenced social media data can have high spatial resolution, allowing researchers to observe spatial patterns in the research areas being examined [34]. Therefore, a second step is often to exclude data lacking relatively high precision location [92] and to exclude geolocated data outside of the study area [31,74]. Gazetteers can also be used to geocode users' locations to latitude/longitude coordinates [65] and thus allow invalid data to be removed. Li et al. [100] suggested that researchers should take into account that not all of the users would like to share their locations when posting messages, thus the data used for analysing UGS are a subset of the entire dataset and the users who include spatial information in their messages are not wholly representative of the entire user base.

More broadly, it is estimated that Twitter's streaming API only released less than 1% of all world-wide generated Tweets [101] and Pew Research Center reported that Twitter users only accounted for about 24% of online adults in 2016 [102], with users more likely to be younger and wealthier than the general population. However, the total number of social media data is very large, so researchers can still obtain great volumes of georeferenced data and attempt to balance these potential sources of bias [100].

Individual social media users have different activity characteristics. Individual Twitter user data, for example, typically have a very long tail; a large proportion of Tweets are produced by only few hundred [100]. In order to remove a similar bias in Flickr data, Pickering et al. [88] suggested capping 10 images per person. In addition to long tail problems, different research aims required specific datasets. For example, Maeda et al. [103] extracted tourists' destinations and generated visitation patterns by using Twitter data and split users into groups of residents and tourists. The sentiment score of geo-Tweets related to UGS in New York was similarly divided into park users and non-park users [72].

### 3.4.2. Methods Used in Spatial Data Analysis

Kernel density estimation (KDE) has been frequently used to quantify the spatial distribution of park visitors across a study area [87,104]. KDE is a statistical approach used to estimate a smooth and continuous distribution from a limited set of observed points [105]. It was used to construct density surfaces from point of interest check-ins [106] and Lee and Tsou [87] used KDE to analyse geotagged Flickr photos, identifying hotspots of tourist behaviours. Han et al. [107] used KDE to explore spatial activity using Twitter, showing that KDE can be used to study the dynamic evolution of georeferenced data across both time and space. Fundamentally, KDE analyses point to the varying distribution of park visitors over fine temporal and spatial scales.

One key variable in the KDE method is the specification of the kernel radius. Adopting different sizes of radii will generate surfaces with different degrees of spatial aggregation or smoothing. Thus, it is important to select a suitable kernel radius when assessing the density of park visitors in urban green space areas. For example, Lee and Tsou [87] examined two spatial scales of KDE for tourist activity analysis. First, 50 km was selected to identify the general regions in the Grand Canyon area, and second, a 200 m kernel was selected to identify smaller hotspots along roads and trails (with a higher spatial resolution).

In addition to the KDE method, K-means, Mean-Shift and DBSCAN algorithms are commonly used to assess the spatial patterns of tourists [22,108]. In order to measure spatial dependence, Moran's $I$ has been used to measure autocorrelation, allowing researchers to explore the degree to which one object value is similar to other nearby object values [31].

### 3.4.3. Methods Used in Temporal Analysis

In terms of temporal analysis, the timestamps of social media contributions have been divided into different temporal categories to trace changes in the number of visitors across the study area [58,69,109]. Such studies analysed the temporal patterns from daily to hourly distributions, weekly patterns to distinguish which parks are more popular at the weekends, and seasonal patterns which reflect the effect of climatic factors. Schirpke et al. [90] and Wakamiya et al. [110] used the same methods to analyse the temporal patterns of outdoor recreation in the European Alps and their surroundings. Spearman correlation coefficients were used to analyse temporal patterns across data derived from different social media data platforms [58].

### 3.4.4. Methods Used in Semantic Analysis

Text mining is very important in social media analysis because it provides the basis for various research objectives including sentiment analysis, emotion detection and topic modelling. Before analysing text data, various preparatory processes must be applied, such as *tokenization* (splitting a sentence into a series of independent words), *stemming* (removing

tenses, capturing singular and plural forms of words) and *structuring* the sentence or text (e.g., "gives", "gave", or "given" are all related to "give"). In addition, some users (and researchers) are not fluent in English and effective translation tools such as Google Translate and iTranslate are needed for addressing problems of language confusion when mining text from Tweets [33].

Sentiment analysis aims to extract opinions towards a topic or events generally from textual data sources and can be applied after text mining to assess the users' emotion and satisfaction in UGS or urban parks. The approach is to compare the stemmed terms to a sentiment lexicon of some kind. For example, SentiStrength V2.2, an opinion mining tool based on a lexicon of words including positive or negative emotion and scores (e.g., happy: 2, bad: −2), was used to investigate sentiments of texts, especially in short texts such as Tweets [92,111,112]. This approach has been proven to achieve high accuracy in sentiment analysis [113]. In addition, word polarity analysis can help researchers calculate the probability of the appearance of the word in a given text [114], which is a good way to extract opinions generally from textual data sources [31]. In the context of UGS, Chapman et al. [115] used three different approaches to investigate the sentiment of Tweets in relation to UGS. The methods were: (1) Manual Annotation, referring to a random sample of 1000 Tweets which were annotated by five annotators—this method provides a robust test set which can be used to compare with other methods; (2) Fully Automated Annotation, referring to an Affective Norms for English Words (ANEW) resource [116], which was used as the basis for emotion annotation instead of manual annotation; and (3) Graph-Based Semi-Supervised Learning Annotation, where the researchers first selected a sample of manually annotated Tweets and then used them to train a graph-based semi-supervised learning algorithm, which was finally used to annotate the remaining Tweets.

A limitation of the previous study is that each message is assigned one kind of emotion. To overcome this, Park et al. [117] classified the sentiment scores of Tweets into three categories: positive (scores 1 to 4); neutral (scores of 0); and negative (scores −1 to −4). Other research has used a similar scoring system, which allows a larger number of tweets to be classified as "neutral", for example, with scores of −2 to 2 [74].

### 3.5. Data Quality Issues and Improvement

VGI has proven very successful as a means of obtaining georeferenced information about social media users at as frequent a time interval as necessary [97]. In addition, these kinds of data can often be freely downloaded via APIs (Table 3), enabling researchers to analyse UGS use at a very low cost. However, VGI has some obvious limitations.

In order to assess the extent to which scholars can rely on Twitter, some researchers have investigated how much information is spam [118]. They found that the high volumes of spam made it difficult to generate useful and meaningful information. Hence, in order to improve the quality of this type of text-based VGI data, it is important to pre-process the social media data before further analysis (as described in Section 3.4.1.) to filter out spam [67], identifying the data within study areas [34], restricting the number of Tweets from prolific users [88], and identifying groups of users, such as urban residents and tourists [72].

For image-based VGI data, different types of smart phones and GPS devices may cause various accuracy errors. For example, georeferenced social media data collected from the web application Wikiloc may lead to uncertainty in data quality [29]. Therefore, although the photographer may usually be relatively close to the subject of the photo, especially in a UGS, and likely within the geolocation error margin, the geolocations of photographs have been found to be influenced by users who prefer to geotag the photo with the location of the photo subject (e.g., a famous building) rather than the photographer's position [94]. Similarly, users who are not familiar with the function of adding geolocations for photos or lack enough spatial knowledge sometimes incorrectly geotag their photographs. Study results can also be biased by users posting many photos from the same location. This problem should not be ignored and some studies have taken steps to remove this bias [29].

In order to improve the locational quality of image-based VGI data, some research has set up a series of 200 m sided hexagons, in which the pictures were aggregated ("binned") and the number of users and photographs was calculated [119]. Under this method, the modifiable-areal-unit-problem (MAUP) effect can be minimised [119]. Similar studies have also applied this approach to analyse data at the user level [120]. The number of photos was capped at 10 images per person in order to remove the bias from a few visitors who post lots of images [88]. Researchers may also want to consider manual image classification when analysing the content of images. For example, the content of an image was initially interpreted by two people, then a third person cross checked the final interpretation and any discrepancies [88].

In terms of map-based VGI data, the lack of common standards across platforms and access to accounts for providing and uploading data may further influence the accuracy of data or user attributes [121]. In addition to accuracy, data completeness also exerts an obvious influence on providing reliable services [122]. GPS tracking applications such as Strava, MapMyFitness, and Wikiloc can provide metadata that contain information about physical activities that park users participate in. This allows researchers to detect the mobile patterns of visitors in park areas [123]. However, GPS tracking data may contain gender bias as men have been found to be more likely to record their activities than women on some applications [124].

To improve data quality, OSM and authoritative data should be combined to develop an integrated open data source [25]. Levin et al. [89] presented a semantic analysis to improve data classification, enhancing data quality to overcome cross-cultural and multi-language problems. Some studies have focused on procedures to enhance quality during the acquisition and compilation steps via crowd-sourcing, social, and geographic approaches [125].

The evaluation of data validity, accuracy, representativeness, and uncertainty is essential when such data are used to analyse UGS visitation patterns and user behaviours [70,107]. In order to evaluate and improve the representativeness of different social media data sources, Blank and Lutz [84] evaluated six platforms including Facebook, LinkedIn, Twitter, Pinterest, Google+, and Instagram in Great Britain. Their results showed that Twitter users tend to be younger and more highly educated. In terms of image-based data, the population representativeness of Flickr was assessed, and users represent a specific subsample of visitors to any site with specific motivations to take and share images, hence Flickr represents only a fraction of the actual visitors [87]. Twitter data have been widely used in UGS research, and some studies [72,118] have suggested that geolocated Twitter data in metropolitan cities can be used as an alternative source of information able to adequately characterise commercial, leisure, and residential areas for urban planners, especially in combination with their geographic location marking and time stamping functions including real-time.

## 4. Discussion

VGI data have been widely used in the research field of UGS analysis. The growing popularity of social networks and social media services has attracted researchers from various disciplines, and this new form of geographic data has been used in a variety of applications. This review has identified the ten most frequent topics from the reviewed articles, with the most common topic related to *cultural ecosystem services*. This study manually extracted research themes across all selected articles which may be influenced by authors' personal views and knowledge, which was a limitation of this review. Various social media platforms have been used as data resources for different objectives in the reviewed articles. The top five popular social media platforms were Twitter, Flickr, Instagram, Weibo and OpenStreetMap, with Twitter and Weibo providing text-based data, Flickr and Instagram providing image-based data and OpenStreetMap providing map-based data. This review also examined a number of geospatial methods used for data collection and analysis, and

highlighted a number of quality issues and suggested methods for improving data quality from the reviewed articles.

*4.1. Research Gaps and Opportunities*

There are many potential areas for further research that have been highlighted by the process of undertaking this review. These relate to the limitations of social media, as identified in this review, including data acquisition, data representativeness, privacy concerns, data quality, as well as differences across social media platforms. Some of the key research gaps and opportunities in the use of social media data in UGS studies are as follows:

- Using data from multiple sources

Much of the previous research has used only a single data source or platform, which may result in a biased representation of the target population and fail to capture the important characteristics of that population [60,72,86]. Twitter has established a new generation of API (Twitter API 2.0), and academic researchers can then collect the full history of public Tweets via Twitter Academic Research API—this provides researchers with a window into understanding the use of Twitter and social media [126]. However, most platforms offer only limited data access to researchers, and the sampling algorithms for platform APIs remain unknown [127]. For example, Wang et al. [68] used the data that were collected from a social media platform Dazhongdianping (www.dianping.com) (accessed on 3 March 2021), which is a website allowing people to provide reviews on local services across China, to assess park use in Beijing and recommended that further analysis should be taken using different data. In other studies, Flickr was used as a sole data source [87]; however, recent changes to the Flickr API and terms of service have caused difficulties in accessing data. Different platforms can provide data describing different aspects of the same place, whereas using only a single platform may cause biases and uncertainties. Comparisons with different kinds of social media platforms and on-site surveys will help improve the generalisability of the studies. An example of an approach that combines multiple sources is that three platforms (Flickr, Panoramio, and Geograph) were used to detect cultural ecosystem services [75]. Their results show different photo sharing behaviours, with Flickr and Panoramio having almost interchangeable results whereby Flickr places greater emphasis on human-made cultural artifacts. A further extension is possible through recent developments in image analysis, which support the automated classification of photographs into known categories, which could be extended into typical UGS features. Such data would enhance the analysis of social media data, especially in the context of examining the features that are most attractive to UGS users and shared across media platforms.

- The need for combining personal information with data analysis

Information about individual users, including gender, age, occupation, and income, is very meaningful for the study of cultural service perception, park use assessment and UGS planning [75]. Whilst recognising that some park users may be reluctant to disclose their personal sensitive information, such as income and sexual orientation, such data may allow a more refined analysis of attitudes and perceptions and may provide confounding or modifying factors in an analysis. Analysing this type of user data is an important part of understanding the variations in perceived information. The lack of such data is not conducive to the subdivision of research data, but can be inferred from the exploration of user posting histories [128]. It may be more effective to combine survey data which may cover more comprehensive individual information to supplement the research results. Only two studies [79,97] used both survey data and social media data in UGS research. In addition, the number of visitors to park areas needs to be accurate as much as possible, as these data can be used to validate the results from social media data and help researchers to comprehensively understand park use. In order to estimate the actual number of park visitors, counters could be set at some parks—this will give accurate data about the number

of people who visit parks. In addition, some municipalities provide free Wi-Fi hubs inside parks and data from these hubs could be used to estimate the number of visitors. These types of data can be used as a complement to questionnaires and social network data methods.

- Improving information mining analysis and models

In order to improve the accuracy of language translation, there are a number of opportunities for more nuanced analyses of social media such as Twitter. Domain-specific lexicons [33] need to be developed specifically for green spaces. In order to generate a more accurate analysis of visitor opinions in social media, future research should consider developing specific, bespoke lexicons for parks, forests, lakes or other related venues as has been done in other domains [129]. In addition, there still exists the challenge of analysing and translating polarity related to negative or positive perception in sentence-level sentiment analysis. For data analysis, there are various methods associated with different kinds of social media data used to analyse UGS. Specifically, in terms of text-based data such as Twitter data and Weibo, it is important to process text-based semantics for sentiment and similar analyses. The analysis of geotagged social media data requires methods to detect the accuracy of the location information [86], and analysis models and workflows need to be further refined. For example, it is difficult to tell whether people mention the Bird's Nest and the Water Cube in Olympic parks because they are attractive or simply to use them as a location reference [68]. Thus, a stronger unsupervised selection technique is needed to analyse these unlabelled, unstructured and inherently linked datasets online. A further improvement to analyses of social media would be to examine the networks suggested by social media posts that are shared, re-posted or reacted to. Here, classic graph theoretical approaches could be used to infer connections, influencers' opinions and spatio-temporal trends in social media data [130]. This is a hugely under-developed area of research that has yet to gain traction in domain-specific analyses of social media such as UGS. Examining such interactions can indicate topics of particular interest and potentially deal with data sparsity issues.

- The representativeness and validation of social media data in UGS research

The representativeness of social media data sources such as Twitter has attracted more and more attention from scholars. For example, British Twitter users tend to be younger, wealthier, and better educated than the general population [84]. However, when research is limited to urban areas, georeferenced Tweets or photos can produce valuable and useful knowledge due to the high density of social media users [72]. It is important that researchers assess the validity of social media data before analysis. For example, Twitter data on park use were validated in Barcelona and Madrid by comparing different data sources including census and cell phone data [85]. The results showed that the three data sources provided comparable information in studies of urban human mobility. Twitter data have been widely used in urban green space research, and some studies [118] have suggested that geolocated Twitter data in metropolitan cities can be used as an effective tool to characterise commercial, leisure, and residential areas for urban planners. Validation can also be through official data such as contemporary census data or survey data provided by local managers. A further dimension to the issue of representativeness relates to general social media usage. A key area of future work is to examine the context of social media analyses using related data to explore whether the use of social media in relation to UGS is correlated to social media usage generally (for example, ease of access), to local cultural social media usage customs or even to the amount of UGS.

### 4.2. Analysis Methods and Approaches

Previous studies analysed VGI data from the aspects of spatiotemporal patterns of data points, text mining and semantic analysis. However, VGI data cleaning and pre-processing play an important role in whole research works.

Researchers should carefully clean the collected datasets before analysing them. For example, social media data such as Tweets can be posted by bots or spammers instead of actual Twitter users, and this may cause data bias and over representativeness, and the sentiments of Tweets can also be overestimated by Tweets that were posted by retailers, job advertisements and shopping malls. More advanced cleaning methods should be used according to different objectives of research works. For example, some studies [72] focused on the differences between park visitors and non-park users, thus, it is important to distinguish the users' categories before analysing the datasets.

As for spatial pattern analysis, this review mainly summarised the KDE as the method which was frequently used in previous studies [87,104,105]. The key issue in using this method is to determine the kernel radius when assessing the density of data points in study areas. In addition to KDE, K-means, Mean-Shift and DBSCAN algorithms are commonly used to assess the spatial patterns of tourists in some studies [22,108]. The approaches that combine different spatial analysis methods should therefore be developed in future works related to UGS research using VGI data. In temporal analysis, different time scales have been used in previous studies that mainly focused on daily, weekly, and monthly visitation patterns. The combination of spatial analysis and temporal analysis could be undertaken in more specific analyses such as at the individual level. For example, a discretised spatial–temporal probabilistic distribution can be used to characterise the Twitter users who posted georeferenced tweets when visiting UGS areas [131]. Further, previous studies mainly analysed UGS visitation to understand the current or past states of UGS use, and few studies have paid attention to the prediction of UGS visitation—future research could focus on the prediction of the UGS visitation mode, especially for holidays such as Christmas and Easter.

Text mining is very important in social media analysis because it provides the basis for various research objectives, including sentiment analysis, emotion detection and topic modelling. This review summarised sentiment analysis methods such as SentiStrength V2.2 [92], word polarity and Graph-Based Semi-Supervised Learning Annotation [115]. In the sentiment classification of texts from Tweets, for example, it is possible that each Tweet contains more than one kind of emotion or sentiment, thus it is important to determine the overlaps amongst different sentiment categories when classifying the sentiments of Tweets. Topic detection also plays an important role in text mining. However, topic detection from unstructured data such as Tweets is challenging due to the short and unstructured content and dynamic environment. Recently, methods used to estimate topics from social media platforms include Latent Semantic Analysis (LSA), Probabilistic Latent Semantic Analysis (PLSA), Nonnegative Matrix Factorisation (NMF) and Latent Dirichlet Allocation (LDA) [132]. The key point of topic modelling for social media data will be combining more text, considering social features and taking the temporal aspect into account as a user's environment always changes in real time. In addition, the number of topics and model selection also play an important role in topic modelling. Future research should take care in selecting suitable and appropriately sensitive approaches for detecting topics in different data sources.

## 5. Conclusions

This paper makes a novel contribution by comprehensively reviewing the scientific literature of research using VGI and social media data to understand UGS. Snowballing [42] was used to capture relevant papers that were not part of the original search but were referenced within the identified papers, and personal knowledge of the literature was used in addition to the systematic search. As such, the literature search is not entirely replicable, which is a limitation. However, it follows well-understood standards for narrative reviews [133]. The variation in the usage of different data platforms has been described and a number of research areas using these data sources have been discussed, as well as data analysis methods and data quality issues in the context of UGS research. A number of limitations associated with social media data were identified in relation to

their coverage, data quality, and representative uncertainties. Researchers using such data should pay particular attention to these, especially in the context of spatial or locational research. Social media data can be cross-validated or linked to other data to overcome the limitations of using data from a single platform, and combining data sources and types in this way allows some of the limitations to be overcome.

There are a number of opportunities for future research, including the need to evolve methods that have a greater analytical depth beyond sentiment and text mining in order to increase the depth of information that is extracted from social media data, for example, linked to preferences and behaviours. In the specific case of urban green space, future research should focus on factors related to physical activities in UGS areas, urban park use and accessibility, all which can be captured from social media data. For example, researchers could determine the motivations of contributors to social networks in sharing UGS-related text and images, and this has the potential to inform on the specific UGS qualities that are being shared (i.e., park accessibility, design configuration, presence of water, etc.). The automated classification of images posted online also has considerable potential. While some research exists regarding motivations and psychological reasons as to why people share (e.g., a personal cause), further research is needed to determine why a certain UGS feature has been shared, the timing of the shared post, the novelty of the content, etc. In addition, there is a need to assess the usability of social media data analysis in public departments involved in decision making processes around UGS. In terms of data analysis, future research should examine approaches that combine textual, image and map data to extract more representative information for UGS. This would require tools to be developed to do this. Overall, social media data are best used with other data sources to gain full and dynamic geotagged images and text on an urban green space issue, for the benefit of people and living quality.

**Author Contributions:** Conceptualization, Nan Cui and Alexis Comber; methodology, Nan Cui and Alexis Comber; software, Nan Cui and Alexis Comber; formal analysis, Nan Cui; literature review and investigation, Nan Cui; writing—original draft preparation, Nan Cui; writing—review and editing, Victoria Houlden, Nick Malleson and Alexis Comber; visualization, Nan Cui; supervision, Alexis Comber, Nick Malleson and Victoria Houlden; funding acquisition, Alexis Comber and Nick Malleson All authors have read and agreed to the published version of the manuscript.

**Funding:** This study was supported and funded by the University of Leeds and the Chinese Scholarship Council (201906390033), and the European Research Council (ERC) under the European Union's Horizon 2020 research and innovation programme (grant agreement No. 757455).

**Institutional Review Board Statement:** Not applicable.

**Informed Consent Statement:** Not applicable.

**Data Availability Statement:** Not applicable.

**Acknowledgments:** We thank the anonymous reviewers whose comments and suggestions helped improve and clarify this manuscript.

**Conflicts of Interest:** The authors declare no conflict of interest.

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
