# Peer review of "Using VGI and Social Media Data to Understand Urban Green Space: A Narrative Literature Review"

_ijgi, doi:10.3390/ijgi10070425_

Round 1

Reviewer 1 Report

Thank you for the opportunity to review the Using VGI and social media data to understand urban green space: A narrative literature review article.

The subject is of interest especially since social media platform usage is fairly ubiquitous in urban environments.

The paper is well written and structured. I would like to see suggestions of more concrete directions of research development, especially related to improving information mining analysis models, spatial and temporal analysis of data, or expanding the lexicon related to UGS. Also people in the users’ network react to the users’ posts, are those reactions considered in these analyses? Do you see any usefulness in user’s network response to the posts? For image platforms, the photos can be automatically classified in well established categories using tensorflow or any other machine learning methods. This classification can enhance the analysis, especially if the interest is in looking at what features are the most attractive to users of UGS and are most shared on media platforms. If possible, as a supplementary material, I would like to see a list of the papers considered in this study, especially since the selection itself cannot be really replicated.

For other comments and suggestion please see the manuscript. I am recommended this paper for publication with minor revisions.

Thank you for the opportunity to review this paper.

Author Response

Dear reviewer, 

Thanks very much for your helpful comments and suggestions, please find attached a point-by-point response to the comments and concerns.

Best regards, 

Nan

Reviewer 2 Report

Comments to authors

The review is very important because it addressed new research area.  There will be a large audience interested in the topic. The review covered major issues related to application of social media and its use as a source of data. The methodology, data processing, and data presentation are very clear. The article includes comprehensive list of references.

The need for combining personal information with data analysis

  1. Counters could be set at some parks: This will give accurate data about the number of people who visit parks. Respondents to a Surveys/questionnaire may be not the same as those who actually visited the parks.
  2. Some municipalities provide Free Wi-Fi hubs inside parks (wireless router). Data from these hubs could be used to estimate number of visitors.  

Research gaps

There is a need to conduct more research on why many people do not like to share their locations. In addition to what is listed related to gender, there may be other reasons such battery.  

There is a need to assess the applicability/usability of the social media by public departments in decision making process. In some cases, 70-80% of social media data is of personal nature and governmental departments find difficulty to get use of it.  

References

The authors need to follow the reference style of the journal

Author 1, A.B.; Author 2, C.D. Title of the article. Abbreviated Journal Name Year, Volume, page range.

In the majority of references cited there is no Journal Name.

Books and Book Chapters:

Author 1, A.; Author 2, B. Book Title, 3rd ed.; Publisher: Publisher Location, Country, Year; pp. 154–196.

Author Response

Dear reviewer,

Thank you very much for your helpful comments and suggestions, please find attached a point-by-point response to the comments and concerns.

Best regards,

Nan

Reviewer 3 Report

This paper reviews the use of VGI and social media data in research examining UGS based on the analysis of 177 papers. It summarizes the characteristics and usage of data from different platforms, provides an overview of the research topics using such data sources, and characterize the research approaches based on data pre-processing, data quality assessment and improvement, data analysis and modeling. A number of important limitations and future priorities are stated. The research topic is interesting but some things need to be clarified and additional information should be provided.

Line 44-47: Authors seem to confuse social media and VGI. VGI is not based on the social media technology as stated in Line 46. Georeferenced data provided by social media can be considered as VGI and social media are VGI sources.  Please check See, L, Estima, J, PÅ‘dör, A, Arsanjani, J J, Bayas, J-C L and Vatseva, R. 2017. Sources of VGI for Mapping. In: Foody, G, See, L, Fritz, S, Mooney, P, Olteanu-Raimond, A-M, Fonte, C C and Antoniou, V. (eds.) Mapping and the Citizen Sensor. Pp. 13–35. London: Ubiquity Press. DOI: https://doi.org/10.5334/bbf.b. License: CC-BY 4.0

A list of all the 177 papers reviewed should be provided. I believe that only some of them can be found in the Reference section.

Maps in Figure 2 should be enlarged as they are too small.

Additional info for MCA need to be provided on the transformation of categorical data to numbers. Regarding MCA, the grouping in Cluster 2 of social media platforms and themes such as ecosystem services, tourism etc need to be elaborated as the statement that “these social  media platforms were selected as the main data sources in this field” is true for topics in Cluster 1 as well. It is not clear what do the results of the MCA contribute to the research. Is the graph in Figure 2b related to MCA?

Graphs in Figure 2a and 2b are too small as well.

Line 194: Table 3?

Line 220: OSM is not a social media platform!!!

In Figures 4 and 5, too many color values appear in the legend and it is difficult to understand the ones appearing in the graph and interpret the values. It is advisable to provide a small color hue schema with 4-5 different hue values and to group the data values in 4-5 categories. Additionally, it would be nice to have statistics for all the papers in the same graph  with the data appearing in lines 210 -214 / 311 – 318.

Manual extraction of phrases for commenting on research themes is rather subjective and may be influenced by the authors’ personal views. This is a limitation of the research and need to be commented in the discussion session.

Good Luck in publishing the paper!!!

Author Response

Dear reviewer.

Thank you very much for your helpful comments and suggestions, please find attached a point-by-point response to the comments and concerns.

With best regards,

Nan

Round 2

Reviewer 3 Report

The paper has been revised and can be published.